# Changes of 25(OH)D Concentration, Bone Resorption Markers and Physical Performance as an Effect of Sun Exposure, Supplementation of Vitamin D and Lockdown among Young Soccer Players during a One-Year Training Season

**DOI:** 10.3390/nu14030521

**Published:** 2022-01-25

**Authors:** Joanna Jastrzębska, Maria Skalska, Łukasz Radzimiński, Guillermo F. López-Sánchez, Katja Weiss, Lee Hill, Beat Knechtle

**Affiliations:** 1Department of Pediatrics, Diabetology and Endocrinology, Gdansk Medical University, 80-210 Gdansk, Poland; joanna.jastrzebska@hotmail.com (J.J.); mariajastrzebska@gumed.edu.pl (M.S.); 2Department of Health and Natural Sciences, Gdansk University of Physical Education and Sport, 80-336 Gdansk, Poland; lukaszradziminski@wp.pl; 3Department of Public Health Sciences, School of Medicine, University of Murcia, Espinardo, 30100 Murcia, Spain; gfls@um.es; 4Medbase St. Gallen Am Vadianplatz, 9001 St. Gallen, Switzerland; katja@weiss.co.com; 5Department of Pediatrics, Division of Gastroenterology and Nutrition, McMaster University, Hamilton, ON L8S4L8, Canada; hilll14@mcmaster.ca; 6Institute of Primary Care, University Hospital Zurich, 8006 Zurich, Switzerland

**Keywords:** seasonal variation, blood parameters, training load, home isolation, COVID-19

## Abstract

The main purpose of this research was to demonstrate the changes in 25(OH)D concentration, bone resorption markers, and physical fitness along the one-year training season in young soccer players. A total of 24 young soccer players (age: 17.2 ± 1.16 years, mass: 70.2 ± 5.84, height: 179.1 ± 4.26 cm) were tested at four different time points across one year (T1—September 2019; T2—December 2019; T3—May 2020; T4—August 2020). After T2 (during COVID-19 lockdown), players were divided into a supplemented (GS) group and a placebo group (GP). Variables such as 25(OH)D, calcium (Ca), phosphorus (P), parathyroid hormone (PTH), aerobic capacity, speed, and explosive power were measured. Analyses performed for all participants indicated significant changes in all selected blood markers and running speed. The highest values in 25(OH)D were noted during summertime in T1 and T4. After individuals were split into two groups, a two-factorial ANOVA demonstrated a significant time interaction for 25(OH)D, Ca, P, PTH, 30 m sprint, and counter-movement jump. Significant time x group effect was calculated for aerobic capacity. This study confirmed that 25(OH)D concentration varies between four seasons, with the greatest decreases in the low sunlight periods. Vitamin D supplementation did not cause a preventive and long-lasting effect of increasing the 25(OH)D concentration in the young soccer players.

## 1. Introduction

The influence of vitamin D on physical health has been investigated over recent years [1,2,3]. Vitamin D is found in bones, kidneys, heart, brain, pancreas, intestines, muscles, and skin [4,5]. It regulates cell maturation, differentiation, and apoptosis [6]. These features are related to the expression of antiapoptotic proteins that could reduce the rate of proliferation [6]. The anti-inflammatory effects of vitamin D and its preventive role in cancer development was previously described [6,7].

Maintaining the optimal level of 25(OH)D concentration in the blood plasma is difficult from autumn to early spring for people living in the countries of northern Europe. Several authors from Poland [2,8,9,10], Russia [11,12], and Great Britain [13] have previously described this issue. These authors reported a significantly lower 25(OH)D concentration in athletes and untrained people during winter than during summer months [13]. Systematic monitoring of bone resorption markers such as calcium (Ca), phosphorus (P), and parathyroid hormone (PTH) during vitamin D supplementation has been suggested, even when low blood levels of 25(OH)D have been observed [13].

The continuous seasonal monitoring of the 25(OH)D concentration in athletes is an important research topic. Quadri et al. [14], after testing over 600 athletes in Switzerland, reported that 25(OH)D concentration was significantly higher during summer than during winter months. Moreover, they suggested that indoor sports athletes are at a higher risk of vitamin D insufficiency. Similar conclusions were presented by Maruyama-Nagao et. al. [15], who compared the results of female athletes training indoors and outdoors. The amplitude of changes in 25(OH)D concentration was larger in women training indoors [15]. They also found that seasonal fluctuations in serum calcium levels were not statistically significant, and parathyroid hormone (PTH) was inversely related to 25(OH)D levels [15].

Few available studies have considered the effect of vitamin D on metabolic changes in young athletes. Especially rare were studies investigating the seasonal changes of 25(OH)D concentration in soccer players [7,11,12] regarding training periods and applied training loads [16]. These studies suggested that the influence of training loads on changes in the parameters characterizing players physical performance was significant. Therefore, when designing experiments in which vitamin D supplementation is used, the same training loads should be used in supplemented and non-supplemented soccer players [16].

Previous studies confirmed the influence of high and low levels of 25(OH)D concentration in athletes blood plasma on the functioning of skeletal muscles [17,18,19], bone resorption [10,13], or the risk of injury [20]. Some authors [20,21,22] observed that frequent damage of muscles and the osteoarticular system in athletes, from countries with a low sunlight exposition, was due to large 25(OH)D insufficiencies.

Although the influence of vitamin D on the level of physical fitness has been previously investigated, the obtained results provided ambiguous results. Numerous authors have suggested that an optimal (>30–50 ng/)mL or high level (>50–100 ng/)mL of 25(OH)D could have a moderate effect on improving the athlete’s aerobic potential [23,24,25,26] and a significant effect on explosive power, running speed [27], strength [28], and the neuro-muscular system [25]. In contrast, both Książek et al. [29] and Michalczyk et al. [9] demonstrated no significant correlations between muscle strength, maximal oxygen uptake (VO_2_max), and vitamin D in professional Polish soccer players during wintertime. Similarly, no significant relations between functional muscle strength and 25(OH)D concentration were detected in professional soccer players [28] and physically active young males and females [30].

Due to the nature of professional soccer, players are at particular risk for vitamin D insufficiency. The reason for this is the large number of official matches played during the annual training cycle where the difference between the pre-season and competitive periods has become indistinct and the detraining period sometimes last less than two weeks [31]. The periodization was more widely observed and respected in training periods of young soccer players [31]. However, the duration of the competition season has been lengthened to include more matches, leading to a collateral decrease in the non-competitive periods. The competition season (from August to December) should include numerous high-intensity exercises [32]. In some countries such as Poland, the winter preparation period lasts between six and eight weeks (January–February), and the summer preparation period about five to six weeks. The primary goal for coaches during these periods is to improve the players fitness levels by increasing the amount of training drills [33]. Moreover, all the random events, such as injuries, result in a significant decrease in physical capacity [33]. A similar effect was observed during the 2019–2020 season, after an extraordinary off-season period caused by the COVID-19 pandemic [34].

After the World Health Organization (WHO) announced the COVID-19 pandemic in March 2019, most of the countries implemented numerous restrictions concerning outside activities [35]. Therefore, this two-month-long lockdown could have significantly influenced vitamin D synthesis and maintained insufficiencies after the previous winter. The number of studies demonstrating an effect of the COVID-19 lockdown on the physical fitness [35,36] is limited. Hadizadeh [37] suggested that vitamin D supplementation is indispensable for maintaining immune resistance during the pandemic. Furthermore, Jimeno-Almazan et al. [38] showed that physical exercise could have a significant influence in preventing the post-COVID-19 syndrome among young, active people.

Due to the significant role of vitamin D in the exercise metabolism of young athletes, especially in countries of Northern Europe, we designed a study investigating the changes of the 25(OH)D concentration in the blood plasma of young soccer players across the annual training cycle. The primary purpose of our study was to demonstrate the influence of these changes and vitamin D supplementation on the level of players physical fitness and specific bone resorption markers. It was hypothesized that the most significant drop in 25(OH)D concentration would be noted between December and March and that vitamin D supplementation would be one of the factors supporting the effectiveness of the applied training loads.

## 2. Materials and Methods

### 2.1. Participants

Forty elite young soccer players took part in the study. Due to the COVID-19 pandemic and other random occurrences, only 24 players completed the project (age: 17.2 ± 1.16 years, body mass: 70.2 ± 5.84, body height: 179.1 ± 4.26 cm, BMI: 21.9 ± 1.7 kg/m^2^). All participants were members of the same team, competing in the Polish Central Junior League and students at the same private high school (sport class with the soccer profile). A minimum required attendance was set on the level of 85%. A majority of the players (74%) were living in the school dormitory and had homogenous standard diet consisting of a large number of vegetables, fruits, and dairy products, overseen by the club’s dietitian.

One month before beginning the project, no vitamin supplements (especially involving vitamin D) were provided. In addition, during the winter and autumn periods, the athletes wore full-body sportswear due to the low outside air temperatures. Therefore, the lack of UV (ultraviolet) exposure was one of the factors that could reduce the natural vitamin D synthesis.

Injuries and illnesses prevented five players from continuing the research. The rest of the group (*n* = 35) was divided into a supplemented group (GS, *n* = 18) and a placebo group (GP, *n* = 17). The players were randomly assigned to the groups and were not significantly different in 25(OH)D concentration between GS and GP at baseline. This project stage was completed by 11 players from GS and 13 players from GP (Figure 1).

Ethical approval was received from the Local Bioethics Medical Committee in Gdansk (Poland) (KB consent number−26/19). The study was conducted in accordance with the Helsinki Declaration of 2013. Following familiarization and explanation of the study objectives, procedures, and methodology, all participants consented to participate. All the participants or their legal guardians were requested to sign the informed consent form.

### 2.2. Inclusion and Exclusion Criteria

Several inclusion criteria were introduced. All the players were members of one club and performed identical training and homogenous diet indicated by the club dietician. Players were instructed not to take individual vitamin D supplementation directly before and during the experiment. Players who did not participate in at least 85% of training sessions were excluded from the study. A total of 24 players met these criteria and were considered during the final analysis (Figure 1).

### 2.3. Study Design

The study employed a cross-sectional study design. Research was conducted on 40 young soccer players between the middle of September 2019 and the end of August 2020. It was assumed that the amount of sunlight where participants were staying would not influence the synthesis of vitamin D in the period from early autumn to the end of the winter. Moreover, the radiation of sun rays could increase the vitamin D synthesis from the beginning of spring.

During the analyzed period, all the players were performing the same training program involving technical and tactical drills; official and friendly matches; or exercises developing aerobic capacity, speed, strength, and power. Within a one-year training cycle, both the volume and the intensity of the training varied depending on the part of the season. The competition season training process was planned according to the next game and included phases of intentional increasing or decreasing the training load. A typical weekly training load in the competition period contained six training sessions for all players and three for individual training (formations; defender, middle players, strikers) and one league game (12–15 h a week). During the preseason, a large number of general training drills were used along with the soccer specific exercise. The preparation period contained higher training volume and several drills performed with high intensity (Table 1).

### 2.4. Vitamin D Measurement

The study design involved the measurements of 25(OH)D concentration, calcium (Ca), phosphorus (P), and parathyroid hormone (PTH) in blood plasma, as well as tests evaluating the level of aerobic capacity, speed, and explosive power. Additionally, during the project, the number of sunny days and the players diet were recorded. All the measurements were performed at the following four times: middle of September 2019 (T1), middle of December 2019 (T2), beginning of May 2020 (T3, the date was changed from the middle of Mach because of the COVD-19 lockdown), and end of August 2020 (T4). 

### 2.5. COVID-19 Lockdown and Home-Based Training

The COVID-19 lockdown occurred in Poland and other countries all over Europe and the world. From the middle of March 2020 to the middle of May 2020, the tested players were not allowed to perform team training sessions. The home-based training (usually performed indoors) was applied instead. This situation did not allow us to carry out the measurements according to the plan (immediately after completing the vitamin D supplementation). Thus, the T3 tests were conducted as soon as possible in the middle of May 2020. The fourth set of measurements was performed as planned at the end of August 2020 (Figure 1).

### 2.6. Procedures

#### 2.6.1. Degree of UV Radiation

The research was conducted in Poland (Gdynia, 54.50° N 18.55° E, 33 m above sea level) in the northern part of Europe. The degree of UV radiation during the project duration, i.e., between September 2019 and August 2020, was recorded on the basis of historical data from the weather online service for the city of Gdynia [39] (Figure 2).

#### 2.6.2. Determination of 25(OH)D, Calcium (Ca), Phosphorus (P), and Parathyroid Hormone (PTH) Concentration in Blood Plasma

The analysis of biochemical blood parameters was carried out in cooperation with the local analytical laboratory. Fluorescent flow cytometry technology used the Sysmex analyzers (Sysmex Europe GmBH, Norderstedt, Germany). The serum concentration of 25(OH)D was assessed with chemiluminescence (CMIA) (Liaison XL, DiaSorin, Saluggia, Italy). The intra-assay CV (coefficient of variation) of the method was 2.4–6.4%, with respect to the range 0–20 ng/ mLdeficit, >20–30 ng/ mLsuboptimal concentration, >30–50 ng/ mLoptimal concentration, >50–100 ng/ mLhigh, >100 ng/ mLpotentially toxic, >200 ng/ mLtoxic [1]. The calcimetry was used to calculate the total calcium (Ca) in the blood serum (Roche Cobas 6000, Roche Diagnostics, Mannheim, Germany). Phosphorus in blood serum was calculated using a spectrophotometric method (Roche Cobas 6000). The intra-assay CV of the method was 1.4% with respect to the range 2.7–4.9 mg/dl. Finally, parathyroid hormone (PTH) in blood plasma (EDTA) was calculated by chemiluminescence (CMIA), Elecsys PTH reagent (Roche Cobas 8000, Roche Diagnostics, Mannheim, Germany). Indirect precision was 3.1%. The intra-assay CV of the method was 3.1%, with respect to the range of 15–65 pg/mL.

#### 2.6.3. Calculation of Average Vitamin D Intake

The average vitamin D intake was evaluate using the diet (V.6.0) (Poland, 2018). The content of vitamin D in the products for each player was assessed by the vitamin D calculator. These measurements were performed during the first and last week of the competition and preparation period of the project.

#### 2.6.4. Supplementation of Vitamin D

Due to large drops in the players’ 25(OH)D concentration, the vitamin D supplementation was applied after T2. Therefore, the supplementation lasted from the middle of January to the middle of March (all preseason period). Subjects from the supplemented group (GS) were given the vitamin D bottle (Vigantol Merck), whereas the non-supplemented group placebo (GP) received identical bottles with sunflower oil. All participants were asked to take 10 droplets per day (vitamin D or sunflower oil). The GS group received 5000 IU of vitamin D daily in the morning. Such doses were in line with previous research [24,27]. Players and the person who administered the supplementation were blinded as to the group’s assignment.

#### 2.6.5. PACER (Progressive Aerobic Cardiovascular Endurance Run) Test

The Progressive Aerobic Cardiovascular Endurance Run (PACER) test was applied to assess the level of aerobic fitness. The test was performed on the running track in the indoor sport hall. Thus, the external conditions were identical during each attempt. The test consists of running for as long as possible between two lines, separating 20 m in two directions, running back and forth until exhaustion. In the test, the participant moves from one point to another, changing direction to the rhythm imposed by an audio signal. The speed obtained in the last completed stage for each player was considered as individual maximal aerobic speed, as well as the total distance reached, and it was used to estimate the VO_2_max [40].

#### 2.6.6. Sprint Test

The sprint tests were performed indoors on a running track at 10 m and 30 m. The subjects performed two attempts at each distance, and only the best (the shortest) time was considered. Before the test, the subjects performed a 20 min warm-up involving accelerations. The sprint times were recorded by the set of photocells (Smart Speed, Fusion Sport, Cooper Plains, Australia) positioned at the starting (0 m) and finishing lines (10 and 30 m) at the height of 0.9 m. Each subject started from a standing position, with his front leg situated 0.6 m before the first photocell. The recover periods were 120 s after 10 m and 240 s after the 30 m sprint.

#### 2.6.7. Explosive Power Measurement

Three types of jumping tests were performed to evaluate the level of explosive power. Before the test, all the subjects performed a 15 min warm-up involving jumping exercises. The test consisted of two maximal vertical jumps without (squad jump, SJ) and two counter movement jumps (CMJ). The recovery period between jumps was 120 s. Only the best (the highest) results (in centimeters) were used in the final analysis. Then, after a 5 min break, the athletes made the 10 possibly highest. Finally, the average height of the 10 jumps was calculated.

### 2.7. Statistical Analyses

The distribution of the data was verified using the Shapiro–Wilk test. The analysis of variance (ANOVA) for repeated measures was used to check potential differences between successive tests. The post hoc honest significant difference (HSD) Tukey test was performed to identify which measurements significant differences appeared. Following the grouping into GS and GP, a two-factorial analysis of variance (ANOVA) was performed for repeated measurements to assess the inter-group and intra-group effects. Moreover, the post hoc observed power was calculated. A partial eta square (*p*η^2^) was calculated to determine the effect’s magnitude. The *p*η^2^ was classified as small (≥0.01), medium (≥0.06), and large (≥0.14) [41]. Additionally, the observed power (OP) was calculated. The interclass correlation coefficient (ICC) and Cronbach alpha (α) were determined to evaluate the reliability between all performed measurements. The Pearson’s correlation was calculated to identify eventual relations between the blood parameters and the results of physical measurements. Correlation values were defined as very strong (r ≥ 0.80), moderately strong (r = 0.60–0.79), fair (r = 0.30–0.59), and poor (r ≤ 0.29). The significance level was set at *p* < 0.05. All the analysis were performed using Statistica software version 13.0 (TIBCO Software Inc., 2017 Palo Alto, CA, USA).

## 3. Results

Significant changes in 25(OH)D, Ca, P, and PTH concentrations between the measurements were demonstrated along the annual training cycle. The highest values of 25(OH)D concentration were reported in T1 and T4 at the end of summertime. The shortest time in 10 m and 30 m runs was recorded in T4. These results were significantly different (*p* < 0.05) from T1. The results of the aerobic capacity (PACER, VO_2_max) and explosive power (SJ, CMJ, 10 jumps) did not significantly change across the analyzed period (Table 2).

The changes in all the analyzed variables across the period with the vitamin D supplementation (from T2 to T4) are presented in Table 3. A significant time effect was identified according to the level of 25(OH)D concentration, blood parameters, running time on a distance of 30 m, and CMJ. Interestingly, a significant *group × time* interaction was found for the distance covered in the PACER test. The highest values of 25(OH)D were recorded for both groups in T4. These differences were significant in comparison with T2 and T3.

A significant a negative correlation was found between the 25(OH)D concentration, sprint times on a distance of 10 m (r = −0.32, *p* = 0.006) and 30 m (r = −0.36, *p* = 0.002), and P (r = −0.40, *p* = 0.001). Moreover, a fair correlation between PTH and Ca (4 = −0.31, *p* = 0.008) was calculated as well (Figure 3).

## 4. Discussion

The main purpose of this research was to demonstrate a potential influence of sun radiation, vitamin D supplementation, and home isolation (lockdown) on the level of selected bone resorption markers and the level of physical fitness in youth soccer players during the one-year training season. It was assumed that 25(OH)D concentration would be highly variable across the four seasons. The most significant drops may occur during the low sunshine radiation period (i.e., in autumn and winter) and home isolation, but vitamin D supplementation could be a factor preventing these changes and reducing the negative effect of vitamin D deficiency on physical fitness.

To our best knowledge, this is the first study reporting seasonal changes in the 25(OH)D concentration with reference to their physical performance across one year period in youth soccer players. The hypothesis that the concentration of 25(OH)D varies significantly between the four seasons of the year, with the most significant decreases in the periods of low sunlight in the autumn and winter months being confirmed. Furthermore, the analysis of the presented results indicated that the home isolation, caused by the COVID-19 pandemic during the spring, contributed to the maintenance of a low 25(OH)D concentration in athletes. However, we have not confirmed the hypothesis that vitamin D supplementation of young soccer players will cause a preventive and long-lasting effect of increasing the concentration of 25(OH)D in the players blood plasma. 

The deficiency of 25(OH)D in blood plasma during autumn and winter seasons in Northern Europe has been previously described [42,43]. However, increasing awareness of these communities related to the necessity of vitamin D supplementation caused lower deficiencies in elderly people and adults compared with Southern European countries [44]. This inverse relationship was observed among children and adolescents. In the countries located at latitude 47–60° N, the deficiency of 25(OH)D in the range of 5–20% was reported, while in Southern European countries, insufficiency was between 4 and 7% [44].

The results presented in the current research indicated a fluctuation of 25(OH)D, bone resorption markers, and physical fitness of young soccer players across the four seasons. Moreover, correlations between these indicators could be demonstrated. Furthermore, the influence of vitamin D supplementation during the unexpected COVID-19 lockdown was also reported.

Bikle [4] and O’Neill et al. [44] suggested that seasonal changes in vitamin D concentration were related to its endogenous production dependent on the amount of melanin in the skin binding the UV rays, daytime, season, and latitude. The process of inhibiting vitamin D production is influenced by clothing and sunscreen [45,46]. 

In Northern Europe, the highest UV radiation is registered during summertime (from May to August), and the lowest from November to February [42,47]. The current research observed the highest radiation in June, July, and August (high UV index-6) and the lowest from October to March (low UV index- < 3). Thus, it can be assumed that the vitamin D synthesis was significantly limited during at least six months. In addition, due to the low outside temperature during this time, players wore sport clothes covering the whole body. Valtuena et al. [48], who tested 408 Spanish athletes, presented similar conclusions. In 82% of their participants, a 25(OH)D concentration below 75 nmol/l (30ng/)mL was reported. During winter, the largest deficiencies were reported in athletes performing their training in the indoor facilities [48]. Unexpectedly, there were no statistically significant differences in 25(OH)D concentration between athletes training indoor and outdoor in the summertime [48]. It seems to be obvious that athletes training outdoor receive higher sun exposure that should result in increased 25(OH)D concentration. Nevertheless, during the summertime, most of the outdoor training sessions are performed early in the morning or late at night to avoid high temperatures. Such a practice greatly reduces the UV radiation.

Daily diet is a fundamental factor in replenishing vitamin D resources for people with limited exposure to sunlight [45]. Kuchuk et al. [49] suggested daily vitamin D intake between 800–2000 IU, corresponding to 20–50 μg/day, which is recommended for young people living in Poland. These recommendations are particularly important between September and April when vitamin D deficiency is noted in 73.5–83.2% of the population [50]. In contrast, Cashman et al. [51] showed that the lowest effective doses of vitamin D could be even below 10–26 μg/day to avoid a deficiency (<50 mmol/L) or a deficit (<25 mmol/L). Similar problems with vitamin D levels were identified in Norway by Brustad et al. [52]. These authors found that low values of 25(OH)D were not registered in people who ate a seafood diet (mainly fatty fish). In the Polish population, no high food consumption containing large amounts of vitamin D is noted [53,54].

The daily vitamin D intake of players living in their homes was 165.35 ± 18.52 IU/day in the T1, 179.46 ± 14.42 IU/day in the T2, and 187.56 ± 15.65 IU/day in the T4, while players living in the school dormitory delivered T1 145.21 ± 18.52, T2 155.46 ± 15.47, and T4 163.2 ± 16.43 IU/day, respectively. Due to the short period between T2 and T3 (one month), the daily vitamin D intake in T3 was not presented. These doses were many times lower in comparison with recommendation for the young people from Poland (800–2000 IU/day) from September to April [49,50].

The presented results demonstrated a significant reduction in the 25(OH)D concentration in the players blood plasma after the first part of the project. At the beginning of this period, in the middle of September, optimal values (>30 ng/mL) were noted in 83.3% of the participants, while in the middle of December (T2), only in 12.5%. Moreover, at this time, a deficit of 25(OH)D was reported in more than 25% of the players. However, in the middle of May (T3), the 25(OH)D concentration level increased only by 0.93%. In our opinion, this fact could be explained by numerous COVID-19 restrictions implemented by Polish authorities at this time. An appropriate level of vitamin D was identified as one of the factors that could reduce the risk of infection by SARS-CoV-2 coronavirus [55]. Therefore, the deficiency of 25OH(D) concentration during the pandemic seems to be at particular concern. The highest vitamin D values were reported in T4 (increased by 34.8% compared to T3). This improvement was probably the result of the increased exposure to sunlight during training sessions and players’ free time from mid-May to the end of August (Table 2).

The analysis of the changes in the physical fitness showed a significant improvement in running times at distances of 10 and 30 m (T1 vs. T4). A disturbed periodization of the training could possibly cause the lack of significant improvements. An extraordinary off-season period occurred due to the pandemic lockdown. Typical training sessions were forbidden during this break and competitions were cancelled [34]. After the pandemic lockdown (T3–T4), an increase in training volume and intensity resulted in maintaining aerobic fitness and improving speed in the analyzed team. Similar results in soccer training cycles were reported by Nobari et al. [32] and Jastrzębska et al. [56].

The preparation period started at the beginning of January 2020. Due to the low level of 25(OH)D concentration in the middle of December, the 8 week vitamin D supplementation program was prepared to investigate its influence on the changes in physical fitness and bone resorption markers. When the WHO announced the COVID-19 pandemic on 10 March 2020, a total lockdown was implemented in Poland and most of the European countries [57]. These decisions affected the training program and measurements originally planned in the study design. Players received individual programs of home-based training. The third round of measurements and partial resumption of the soccer training was possible in the middle of May. Supplementation lasted according to the plan and was not continued during the lockdown. After this period, the vitamin D level increased by 9.4% in GS and by 5.3% in GP.

Previous research investigating the changes of vitamin D after the same supplementation procedure in the same seasons demonstrated significant changes in the 25(OH)D concentration in young soccer players [16,24]. Similar effects were observed after a 60 day long intervention with vitamin D supplementation (5000 UI/day) by Bezuglow et al. [11], who found a 92% increase of this variable in young soccer players. The results presented in our paper indicated no significant changes between GS and GP two months after completing the supplementation. Therefore, it can be assumed that the positive effects of this dietary management are short-term, and the administration of vitamin D (from 2000 IU to 5000 IU) should be continued when exposure to sunlight is limited. The UV radiation in April and May should naturally improve 25(OH)D. This assumption was supported by the results of Holick [58], Reicharth [59], and Holick and Chen [60], who suggested sun exposure the basic source of vitamin D synthesis in children and adolescents. During the standard diet, vitamin D products usually do not get more than 800 IU/day, which is necessary to maintain the 25 (OH) D at 30 ng/L [60].

Changes in the physical fitness indicators did not provide a clear answer on whether the 8 week vitamin D supplementation (5000 IU/day) could affect the level of selected components of the players’ condition. Significant negative correlations between the 25(OH)D concentration sprint times on the distances of 10 m and 30 m suggested that players with a higher vitamin D level reached better times in the speed tests. However, no other significant correlations with physical fitness measurements were found. The anaerobic fitness components were highest in T4 in both groups (GS and GP). In GS, significant improvements were noted in 10, 30, and the 10 jump tests, while players from GP improved only their times in the 30 m sprint. It can be stated that the vitamin D resources stored in athlete’s body effectively influence the level of their anaerobic fitness. However, it is still unclear as to whether the observed changes are due to differences in 25(OH)D concentration or applied training loads. This topic was considered by Skalska et al. [16], who suggested that a higher 25(OH)D concentration with the same training load can effectively improve the level of anaerobic fitness. In our opinion, the changes in speed and explosive power demonstrated in the current paper were caused by the applied training loads instead, rather than by changes in 25(OH)D concentration, because in both GS and GP, the optimal levels of this indicator were recorded at the final stage of the project.

Our results are in line with the data presented by Koundourakis et al. [25], Dahlquist et al. [23], and Książek et al. [2,29], who found high correlations between 25(OH)D concentration and indicators of anaerobic fitness components. In our research, higher values of 25(OH)D concentration in T4 corresponded to the shortest sprint times and the highest jumps. In contrast, Bezuglow et al. [11] did not confirm such relations in young soccer players during the winter preparation period. Similarly, Montenegro et al. [30] and Gilic et al. [61] did not observe strength improvements after vitamin D supplementation.

Relations between vitamin D level and aerobic fitness in athletes are unclear. Fitzgerald et al. [62] reported no changes in aerobic capacity of players with higher 25(OH)D concentration. Forney et al. [63] and Jastrzębska et al. [24] found higher VO_2_max values in participants supplemented with vitamin D. Interestingly, in our paper, the level of aerobic fitness (PACER and VO_2_max) in GS significantly decreased at the end of the project. Moreover, the significant *group × time* interaction calculated for distance covered in PACER test suggest that vitamin D supplementation did not affect positively the players’ aerobic capacity.

Another important aspect of 25(OH)D concentration variability are simultaneous changes of such bone resorption markers as Ca, P, and PTH. According to Kopeć et al. [8] and Galan et al. [64], these markers were significantly correlated with 25(OH)D concentration in blood plasma. In line with these studies, a higher level of vitamin D was related to a lower level of PTH. Moreover, a significant decrease in 25(OH)D concentration between T1 and T2 corresponded to a significant increase in Ca and P, as well as a significant decrease in PTH. Regarding bone resorption, this could be considered as beneficial metabolic effect. Excessive secretion of PTH has several effects, as this hormone is a uremic toxin that may adversely affect mineral and bone metabolism [65]. The bone resorption results of players after dividing them into GS and GP seem to be interesting. A low level of 25(OH)D maintained from December to May (pandemic lockdown period) resulted in a significant increase of PTH with a contemporaneous decrease in Ca and P for both groups. This is a negative metabolic effect when considering the indicators regulating bone resorption. At the further stage of the project, after the summertime (T4), a significant reduction of PTH with significant improvement of 25(OH)D concentration and stable values of Ca can be considered as a positive effect.

According to Maïmoun and Sultan [66], physical activity influences bone resorption markers and depends on the type of physical effort. Lombardi et al. [67] found no significant relations between 25(OH)D concentration and bone resorption hormones. These authors stated that changes in these variables were caused rather by applied training load than by changes in 25(OH)D concentration in blood plasma. In the current study, PTH was significantly and negatively related to Ca and P, and positively correlated with 25(OH)D. These non-characteristic changes could be the result of the applied training loads and the extraordinary situation of home isolation.

Despite numerous strengths and novelty delivered in this research, some limitations should be indicated as well. First of all, the research was not conducted in agreement with initial study design. The T3 measurements were originally scheduled for half of March, immediately after the supplementation period. However, due to the COVID-19 lockdown occurrence, it was impossible to complete all the tests. Secondly, the sample size after dividing participants into GS and GP could be considered as a limitation as well. Larger number of involved players would help to increase the power of statistical analysis. Nevertheless, soccer teams usually include no more then 25–30 players. Therefore, increasing the sample size seems to be difficult in such research.

## 5. Conclusions

On the basis of the results presented in our study, it can be stated that the hypothesis concerning changes of 25(OH)D concentration along the season in young soccer players blood plasma was confirmed. The highest values were observed in the summertime, while the lowest were at the end of autumn and winter. We reported that vitamin D supplementation (5000 IU/day) for 8 weeks does not have a long-term effect on its level, and after two months, the benefits are insignificant. The COVID-19 lockdown (home isolation) negatively influenced players’ vitamin D synthesis and development of their physical fitness. Due to an unplanned off-season period, the assumed training loads were not completed. Moreover, we demonstrated that a higher 25(OH)D concentration is a factor that significantly supports players’ speed. Furthermore, detrimental changes in bone resorption indicators were not identified during the annual training cycle. The research was conducted along the whole calendar year in the same group of soccer players and allows for reliable analysis of changes in 25(OH)D concentration, bone resorption markers, and physical activity. Such research should be carried out in the future, involving even larger numbers of participants.

## Figures and Tables

**Figure 1 nutrients-14-00521-f001:**
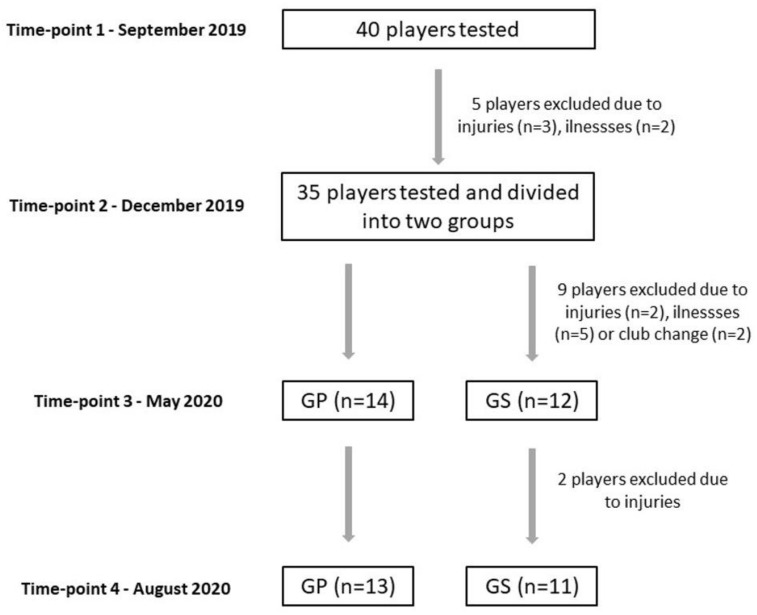
Timeline of the study design and flowchart of the participants.

**Figure 2 nutrients-14-00521-f002:**
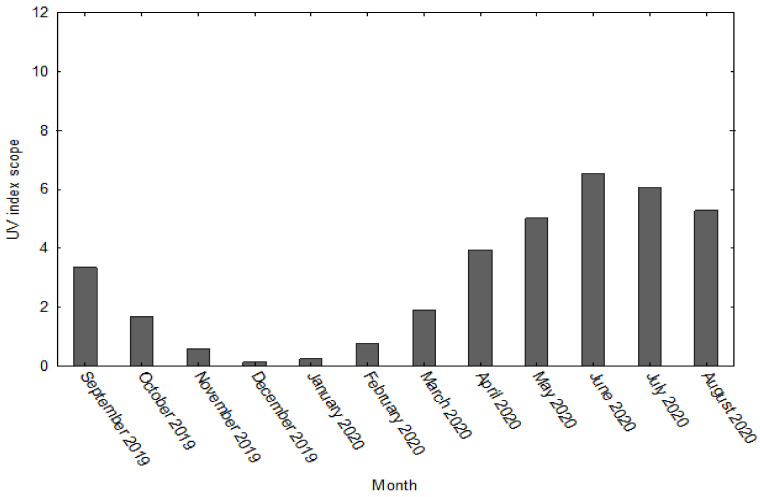
Scope of the UV index for the city of Gdynia (Poland) in the research project [39].

**Figure 3 nutrients-14-00521-f003:**
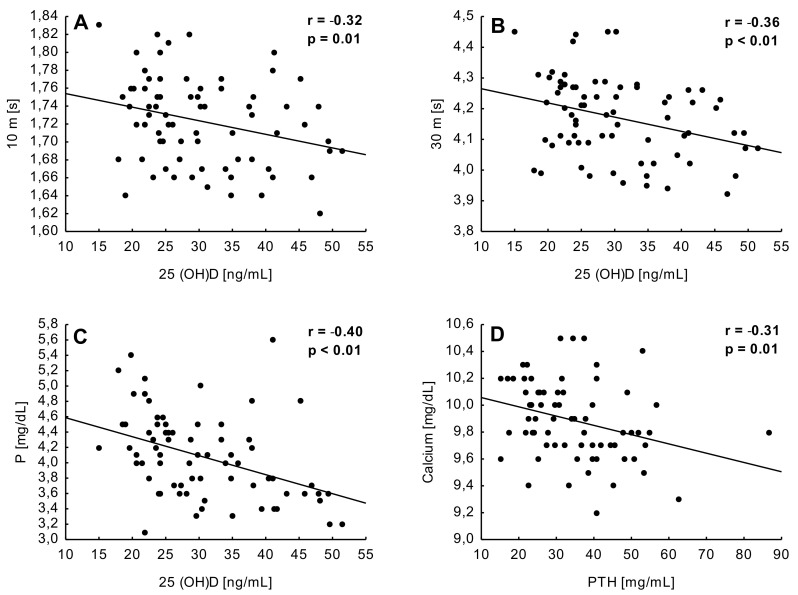
Correlations between chosen blood parameters and sprint test results in tested young soccer players. (**A**) A correlation between 10 m time and 25(OH)D concentration; (**B**) between 30 m time and 25(OH)D concentration; (**C**) between phosphorus (P) and 25(OH)D concentration; (**D**) between calcium and parathyroid hormone (PTH).

**Table 1 nutrients-14-00521-t001:** Overview of the typical weekly training load during competition and preparation periods of the one-year soccer season.

Competition Period		
Day of the Week	Training Drills	
	Morning	Afternoon
Monday	endurance, technical, tactical	Free
Tuesday	speed, technical, small-sided games	individual training (formation)
Wednesday	stretching, regeneration	individual training (formation)
Thursday	plyometric and speed, technical, tactical	individual training (formation)
Friday	coordination, technical, tactical	Free
Saturday	Competition game	
Sunday	Free day	
**Preparation period**		
Day of the week	Training drills	
	Morning	Afternoon
Monday	endurance, technical, tactical	fitness, strength
Thursday	speed, technical, small-sided games	individual training (formation)
Wednesday	coordination, strength	individual training (formation)
Thursday	endurance, technical, tactical	technical, small-sided games
Friday	coordination, tactical	free
Saturday	Friendly game	
Sunday	Free day (regeneration)	

**Table 2 nutrients-14-00521-t002:** Changes in selected biochemical indicators of blood and physical fitness in young soccer players (*n* = 24) during a one-year training cycle.

Time-Point	T1	T2	T3	T4	ICC	α
25(OH)D (ng/mL)	35.0 ± 6.26	24.5 ± 4.89 *	26.4 ± 15.32 *	40.5 ± 6.86 *^,#,^†	0.38	0.81
Ca (mg/dL)	9.5 ± 0.28	10.0 ± 0.24 *	9.8 ± 0.29 *^,#^	9.8 ± 0.30 *	0.62	0.82
P (mg/dL)	3.8 ± 0.47	4.3 ± 0.53 *	4.1 ± 0.52	3.9 ± 0.58	0.73	0.79
PTH (mg/mL)	39.3 ± 15.99	32.4 ± 15.86 *	39.8 ± 10.89 ^#^	32.8 ± 11.06 †	0.81	0.85
PACER (m)	2379 ± 174.9	2473 ± 201.1	2473 ± 241.6	2423 ± 173.6	0.92	0.94
VO_2_max (ml/kg/min)	57.6 ± 2.62	58.7 ± 2.89	58.7 ± 3.35	58.1 ± 2.26	0.92	0.94
10m (s)	1.75 ± 0.05	1.73 ± 0.05	1.73 ± 0.05	1.71 ± 0.06 *	0.81	0.84
30m (s)	4.24 ± 0.13	4.21 ± 0.15	4.19 ± 0.11	4.12 ± 0.12 *	0.89	0.95
SJ (cm)	38.0 ± 3.68	37.9 ± 4.30	37.9 ± 3.47	38.0 ± 3.92	0.88	0.87
CMJ (cm)	45.9 ± 3.30	44.2 ± 4.85	44.6 ± 3.93	45.9 ± 3.65	0.88	0.92
10 jumps (cm)	40.7 ± 3.24	39.4 ± 3.89	39.4 ± 3.29	40.4 ± 2.78	0.85	0.86

* Significantly different from T1; ^#^ significantly different from T2; † significantly different from T3; Ca—calcium; P–phosphorus; PTH—parathyroid hormone; SJ—squat jump; CMJ—counter movement jump; ICC—interclass correlation coefficient; α—Cronbach’s alpha.

**Table 3 nutrients-14-00521-t003:** Changes in selected biochemical indicators of blood and physical fitness in young soccer players in the supplemented group (GS) and placebo (GP) in the T2, T3, and T4 time points of the research program.

Group	GS			ICC	α	GP			ICC	α	Inter-Actions	*p*	pη2	OP
Time-Point	T2	T3	T4			T2	T3	T4						
25(OH)D (ng/mL)	24.1 ± 5.23	26.6 ± 6.35	41.7 ± 6.95 *,†	0.18	0.76	24.8 ± 4.77	26.2 ± 4.52	39.5 ± 6.91 *,†	0.11	0.69	time	<0.001	0.82	1.00
Ca (mg/dL)	10.0 ± 0.32	9.8 ± 0.30 *	9.9 ± 0.31	0.74	0.79	10.0 ± 0.15	9.8 ± 0.28 *	9.7 ± 0.28 *	0.59	0.75	time	0.001	0.27	0.93
P (mg/dL)	4.31 ± 0.52	4.13 ± 0.57	4.03 ± 0.79	0.78	0.81	4.23 ± 0.55	3.99 ± 0.49	3.83 ± 0.32 *	0.58	0.68	time	0.012	0.18	0.78
PTH (ng/mL)	33.9 ± 11.64	41.0 ± 11.83 *	31.8 ± 7.74 †	0.71	0.81	31.0 ± 19.11	38.8 ± 10.41 *	33.5 ± 13.54	0.77	0.83	time	0.003	0.23	0.88
PACER (m)	2518 ± 145.5	2491 ± 188.8	2380 ± 138.9 *	0.84	0.94	2434 ± 237.4	2459 ± 285.8	2456 ± 196.6	0.90	0.92	group × time	0.011	0.19	0.79
VO_2_max (ml/kg/min)	59.2 ± 1.91	59.0 ± 2.60	57.7 ± 1.57 *,†	0.78	0.91	58.4 ± 3.54	58.5 ± 3.98	58.4 ± 2.73	0.89	0.91	-	-	-	-
10m (s)	1.75 ± 0.04	1.76 ± 0.04	1.72 ± 0.06 *,†	0.56	0.66	1.71 ± 0.05	1.71 ± 0.04	1.71 ± 0.06	0.74	0.78	-	-	-	-
30m (s)	4.24 ± 0.17	4.24 ± 0.11	4.15 ± 0.12 *,†	0.76	0.89	4.18 ± 0.13	4.14 ± 0.10	4.08 ± 0.11 *,†	0.82	0.93	time	<0.001	0.42	0.99
SJ (cm)	36.9 ± 3.59	37.9 ± 3.39	37.3 ± 3.95	0.72	0.80	38.9 ± 4.76	37.9 ± 3.68	38.5 ± 3.95	0.82	0.86	-	-	-	-
CMJ (cm)	43.7 ± 4.91	44.2 ± 3.66	45.6 ± 4.03	0.80	0.88	44.7 ± 4.95	44.9 ± 4.26	46.1 ± 3.44	0.86	0.90	time	0.031	0.15	0.65
10 jumps (cm)	38.8 ± 4.24	39.3 ± 3.82	40.9 ± 3.44 *	0.71	0.82	39.9 ± 3.67	39.4 ± 2.93	39.9 ± 2.10	0.82	0.84	-	-	-	-

* Significantly different from T2; † significantly different from T3; GS—supplemented group; GP—placebo group; P—phosphorus; PTH—parathyroid hormone; SJ—squat jump; CMJ—counter movement jump, pη2—partial eta squared, OP—observed power; ICC—interclass correlation coefficient; α—Cronbach’s alpha.

## Data Availability

The data presented in this study are available on request from the corresponding author.

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
