# Peer review of "Changes of 25(OH)D Concentration, Bone Resorption Markers and Physical Performance as an Effect of Sun Exposure, Supplementation of Vitamin D and Lockdown among Young Soccer Players during a One-Year Training Season"

_nutrients, 2022, doi:10.3390/nu14030521_

Round 1

Reviewer 1 Report

The authors studied the changes of 25(OH)D concentration, bone resorption markers, and physical performance as an effect of sun exposure, supplementation of vitamin D, and lockdown among young soccer players during a one-year training season.

Abstract

  • The abstract needs to contain anthropometric characteristics of participants such as age, height, weight, etc.
  • The p-value should be mentioned for all the significant changes mentioned in the abstract.

Introduction

  • Lines 41-43: Provide at least two references to support the following sentence.

Vitamin D was also found to have anti-inflammatory effects that can delay and prevent the development of cancers [6].

Methods

  • Day-to-day test reliability and intraclass correlation coefficients must be included for ALL the assessments.

Discussion

  • What is the novelty of this study? It needs to be mentioned in the “Discussion”.

Author Response

We would like to thank the Reviewer for valuable comments. We have implemented all the necessary corrections according to your suggestions. The specific comments were addressed point by point below. We strongly believe that modified version of the manuscript will meet the Reviewers demands.

The authors studied the changes of 25(OH)D concentration, bone resorption markers, and physical performance as an effect of sun exposure, supplementation of vitamin D, and lockdown among young soccer players during a one-year training season.

Abstract

  • The abstract needs to contain anthropometric characteristics of participants such as age, height, weight, etc.
  • The p-value should be mentioned for all the significant changes mentioned in the abstract.

Response:

The anthropometric characteristics were introduced in the abstract. Unfortunately, the word count limitation in the Nutrients journal require that abstract can not exceed 200 words. Current version of the abstract contain exactly this number. In this context, adding “p” values in this section would require the removal of other important information. Therefore, we suggest no to mention “p” in the abstract.

Introduction

  • Lines 41-43: Provide at least two references to support the following sentence.

Vitamin D was also found to have anti-inflammatory effects that can delay and prevent the development of cancers [6].

Response:

We introduced another citation after the mentioned statement. Review published by Grant (2020, doi: 10.21873/anticanres.13977) confirm the positive role of Vitamin D in preventing the cancer development.

Methods

  • Day-to-day test reliability and intraclass correlation coefficients must be included for ALL the assessments.

Response:

According to Reviewer suggestions the intraclass correlation coefficients (ICC) test reliability (Cronbach α) were calculated and introduced for all the assessments. These values were presented in Table 2 and Table 3.

Discussion

  • What is the novelty of this study? It needs to be mentioned in the “Discussion”.

Response:

The novelty of the study was added at the beginning of the Discussion section as follows:

“To our best knowledge this is the first study reporting seasonal changes in the 25(OH)D concentration across one year period in youth soccer players. Moreover, these changes were demonstrated with reference to physical fitness variability in four different time-points of the season.”

Reviewer 2 Report

The authors present interesting work about the effect of metabolic changes in the context of vitamin D among young athletes. The study is very informative and adds value to the current body of research, however there are few points that should be addressed.

The English editing is needed as there are few errors like players' in line 182, line 75 suggest to use "provided ambiguous results". In addition, there are some redundant lines, 329-333 in the text.

The conclusion and discussion need more improvement and suggest to explain more why the differences in 25(OH)D concentrations between athletes training indoor and outdoor in the summer were not as significant as in winter since it makes more sense to have the opposite effect. Citing the studies which concurred same conclusion is not supported if explanation is not elaborated.

-The rationale for using 5000 IU to GS should be explained as other researchers used different doses.

-I suggest referring also to additional review papers about supplementation and athletic performance, one informative review is by W. B. Grant Grant WB, Lahore H, Rockwell MS. The Benefits of Vitamin D Supplementation for Athletes: Better Performance and Reduced Risk of COVID-19. Nutrients. 2020 Dec 4;12(12):3741. doi: 10.3390/nu12123741. PMID: 33291720; PMCID: PMC7761895.

-The Study limitations and significance part is completely missing from the manuscript and there are numerous limitations to be listed including the small sample size and other factors.

Author Response

We would like to express our gratitude for the Reviewer effort put into our study evaluation. All the essential corrections were applied in the revised version of the paper. Moreover, all the Reviewer comments were addressed below. We hope that introduced modifications will meet your requirements.

The authors present interesting work about the effect of metabolic changes in the context of vitamin D among young athletes. The study is very informative and adds value to the current body of research, however there are few points that should be addressed.

The English editing is needed as there are few errors like players' in line 182, line 75 suggest to use "provided ambiguous results". In addition, there are some redundant lines, 329-333 in the text.

Response:

Paper received extended editing for English language improvement. However, if Reviewer or Editor finds it necessary we will send the text to professional English corrections service. In addition, all the mentioned by Reviewer suggestions were applied and part of the text indicated as redundant was removed.

The conclusion and discussion need more improvement and suggest to explain more why the differences in 25(OH)D concentrations between athletes training indoor and outdoor in the summer were not as significant as in winter since it makes more sense to have the opposite effect. Citing the studies which concurred same conclusion is not supported if explanation is not elaborated.

Response:

Both, Discussion and Conclusion sections were significantly modified. Some additional explanations were presented in these parts of the manuscript. The issue concerning the 25OH(D) concentration in indoor and outdoor athletes during summertime was developed in the text as follows:

“…Unexpectedly, there were no statistically significant differences in 25(OH)D concentration between athletes training indoor and outdoor in the summertime [48]. It seems to be obvious that athletes training outdoor receive higher sun exposure what should result in increased 25(OH)D concentration. Nevertheless, during the summertime most of the outdoor training sessions are performed early in the morning or late at night to avoid high temperatures. Such a practice greatly reduces the UV radiation.”

-The rationale for using 5000 IU to GS should be explained as other researchers used different doses.

Response:

We agree that in the available literature we can find numerous different propositions of Vitamin D intake. However, this is not the first time that the dose of 5000 IU was applied. Here are some examples of studies where participants received such supplementation: Jastrzębska et al. (2018, doi: 10.2478/hukin-2018-0033), Jung et al. (2018, doi: 10.1123/ijsnem.2017-0412), Close et al. (2013, doi: 10.1080/02640414.2012.733822). We added the justification of our choice supported by previous researches in 2.6.4. section.

-I suggest referring also to additional review papers about supplementation and athletic performance, one informative review is by W. B. Grant Grant WB, Lahore H, Rockwell MS. The Benefits of Vitamin D Supplementation for Athletes: Better Performance and Reduced Risk of COVID-19. Nutrients. 2020 Dec 4;12(12):3741. doi: 10.3390/nu12123741. PMID: 33291720; PMCID: PMC7761895.

Response:

Presented by Reviewer paper provide numerous important findings not only about relations between vitamin D and athletic performance, but also about influence of supplementation on the COVID-19 disease. Thus, we decided introduce this review in the manuscript according to Reviewer advice.

-The Study limitations and significance part is completely missing from the manuscript and there are numerous limitations to be listed including the small sample size and other factors.

Response:

According to Reviewer suggestions, paragraph about the limitations was added in the final part of the Discussion section as follows:

“Despite numerous strengths and novelty delivered in this research, some limitations should be indicated as well. First of all, the research was not conducted in agreement with initial study design. The T3 measurements originally scheduled for half of March, immediately after the supplementation period. However, due to the COVID-19 lockdown occurrence, it was impossible to complete all the tests. Secondly, the sample size after dividing participants into GS and GP could be considered as a limitation as well. Larger number of involved players would help to increase the power of statistical analysis. Nevertheless, soccer teams usually include no more then 25-30 players. Therefore, increasing the sample size seems to be difficult in such research.”

Round 2

Reviewer 2 Report

All comments had been addressed